# Marine Transcriptomics Analysis for the Identification of New Antimicrobial Peptides

**DOI:** 10.3390/md19090490

**Published:** 2021-08-28

**Authors:** Baptiste Houyvet, Yolande Bouchon-Navaro, Claude Bouchon, Erwan Corre, Céline Zatylny-Gaudin

**Affiliations:** 1Biologie des Organismes et Ecosystèmes Aquatiques (BOREA) Muséum National d’Histoire Naturelle, Sorbonne Université, Université de Caen-Normandie, Université des Antilles, CNRS-8067, IRD 207, 75231 Paris, France; baptiste.houyvet@satmar.fr (B.H.); yolandebouchon1@gmail.com (Y.B.-N.); claudebouchon1@gmail.com (C.B.); 2BOREA, Normandie University, University of Caen-Normandy, Esplanade de la Paix, CEDEX, 14032 Caen, France; 3SATMAR, Société ATlantique de MARiculture, Research and Development Department, 50760 Gatteville, France; 4Laboratoire d’Excellence ‘‘CORAIL’’, Université des Antilles, Campus de Fouillole, BP 592, Pointe-à-Pitre, 97159 Guadeloupe, France; 5Plateforme ABiMS, Station Biologique de Roscoff, CNRS-Sorbonne Université, 29688 Roscoff, France; corre@sb-roscoff.fr

**Keywords:** venomous fish, lionfish, antimicrobial peptides, transcriptomic

## Abstract

Antimicrobial peptides (AMPs) participate in the immune system to avoid infection, are present in all living organisms and can be used as drugs. Fish express numerous AMP families including defensins, cathelicidins, liver-expressed antimicrobial peptides (LEAPs), histone-derived peptides, and piscidins (a fish-specific AMP family). The present study demonstrates for the first time the occurrence of several AMPs in lionfish (*Pterois volitans*). Using the lionfish transcriptome, we identified four transcript sequences encoding cysteine-rich AMPs and two new transcripts encoding piscidin-like peptides. These AMPs are described for the first time in a species of the Scorpaenidae family. A functional approach on new pteroicidins was carried out to determine antimicrobial sequences and potential uses, with a view to using some of these AMPs for human health or in aquaculture.

## 1. Introduction

Antimicrobial peptides (AMPs) are the major cornerstone of the innate immune system that protects the host against infection. Their diversity is huge: they are found in all living organisms ranging from microorganisms to pluricellular organisms like plants and animals. They also differ in structure and activity [1,2,3,4]. Fish are a major component of the aquatic fauna. They live in a microbe-rich environment and are permanently surrounded by aquatic pathogens. Their skin has a key function in physical and physiological protection as a first defense barrier where AMPs play an antagonistic role against invasive pathogens. Several AMPs were recently identified in fish, including liver-expressed antimicrobial peptides (hepcidins and LEAP-2), defensins, cathelicidins, histone-derived peptides, and a fish-specific family called the piscidins [5].

Among the AMPs characterized so far in fish, some of them are also found in mammals, in particular cysteine-rich peptides that make up a large AMP group including LEAPs, β-defensins, NK-lysins and cathelicidins. Concerning fish LEAPs, the first hepcidin was described in the gills of hybrid striped bass after bacterial challenge [6]. Mature fish hepcidins can have six or eight cysteine residues, as observed in winter flounder (*Pseudopleuronectes americanus*) and Atlantic salmon (*Salmo salar*) hepcidin-like peptides [7]. Besides antimicrobial activity against fungi and both Gram-positive and Gram-negative bacteria [8,9], hepcidin can act as an iron-regulatory hormone [10]. A second cysteine-rich antimicrobial peptide derived from liver or LEAP-2 was identified in fish, with four highly conserved cysteine residues. The first two genes of LEAP-2 were identified in rainbow trout (*Oncorhynchus mykiss*); they displayed constitutive expression in the liver and inducible expression in the intestine and skin [11]. LEAP-2 is an important immune factor involved in the immune response against pathogens in fish [11,12,13]. LEAP-1 and -2 derived from Antarctic notothenioid (*Dissostichus mawson)* and eelpout (*Lycodichthys Dearborni)* revealed bactericidal activity against Gram-positive as well as Gram-negative bacteria, particularly at low temperatures [14]. Defensins have a conserved compact structure displaying a set of cationic loops and they have multiple innate immune functions [15]. Vertebrate defensins have a characteristic β-sheet-rich fold involving six cysteines connected by disulfide bonds. The family is subdivided into α-defensins and β-defensins based on the pairing of cysteines and the length of the amino acid sequences between the cysteines [16]. Only β-defensin-like structures have been observed in fish to date [17,18,19]. The first fish defensins were discovered by Zou et al. [18] who identified seven β-defensins in the three fish species: zebrafish (*Danio rerio*), fugu (*Takifugu rubripes*) and tetraodon (*Tetraodon nigroviridis*). The biological activity of fish β-defensins has been studied and demonstrated: antimicrobial activity against Gram-negative and Gram-positive bacteria, activity against fish-specific viruses, and chemotactic capacity [17,20,21]. Antimicrobial activity has also been observed for NK-lysins derived from fish [22,23]. NK-lysin was initially identified as an antimicrobial peptide from porcine natural killer cells and cytotoxic T lymphocytes [24]. In fish, several NK-lysins have been identified, with a conserved saposin domain including four cysteines and at the origin of their antimicrobial activity [25].

Most of the cysteine-rich peptides of fish display important sequence homologies, except cathelicidins which display sequence divergences. Cathelicidin precursors contain a signal peptide, a conserved cathelin-like domain with four cysteines, and a variable C-terminal antimicrobial domain with or without cysteines [26,27,28]. Cathelicidins have been identified in several fish; the first peptides were identified in Atlantic hagfish (*Myxine glutinosa*) [29], rainbow trout (*O. mykiss*) and Atlantic salmon (*S. salar*) [30]. The antimicrobial activity of fish cathelicidins is variable. For example, codCATH—a cathelicidin from Atlantic cod (*Gadus morhua*)—is highly active against Gram-negative bacteria and fungi but less active on Gram-positive bacteria [31]. Fish can have several cathelicidin genes with different expression profiles following bacterial challenge. In rainbow trout, the rtCATH-2 gene displays a high basal expression level whereas rtCATH_1 is induced upon bacterial challenge [32]. In ayu (*Plecoglossus altivelis*), expression of the cathelicidin gene has been detected in numerous tissues including gill, liver, spleen and intestine, with a time-dependent induction after challenge [33]. Other AMPs that do not contain any cysteine and with little sequence homology have been identified in fish *e.g.,* piscidins [34]. Piscidins possess 18 to 46 amino acid residues; they are amphipathic α-helical peptides [35,36]. The first piscidin to be identified was pleurocidin, a 25-residue peptide isolated and characterized from *Pleuronectes americanus* [37]. Piscidins are present in a wide range of teleosts, particularly in the order Perciformes [38]. They are localized in gill, skin, intestine, head-kidney and spleen tissues. They are an important component of fish immune defense against pathogens [37], and their antimicrobial activity is strong against a range of microorganisms and maintained under high salt concentrations [39].

Antimicrobial peptides have been studied in many non-venomous teleosts [5]. Only one study has been carried out on a venomous teleost—lionfish (*Pterois volitans*)—so far, and has led to the identification of pteroicidin α, a piscidin-like-peptide [40]. Lionfish recently invaded the western North Atlantic region, and more particularly the Caribbean Sea [41]. This venomous species is native from the Indo-Pacific Ocean. It was accidentally released from a public aquarium in Florida in 1992 and since then has colonized most of the coasts of the Gulf of Mexico and the Caribbean Sea. This aggressive predator represents a major threat to local biodiversity and local coral reef ecology [42,43]. The success of its initial invasion was probably due to the lack of adapted predators, parasites and microbial pathogens. The situation has evolved however, with the emergence of new predators [44,45,46,47], parasites [48,49,50], and diseases [51]. This has resulted in a drop in *P. volitans* abundance in the Caribbean area [52] (personal observations). However, the presence of pathogenic bacteria in lionfish in the natural environment or in captivity in aquaria has not been described to date. The present study focuses on the diversity of antimicrobial peptides present in lionfish. We associated a transcriptomic approach carried out from annotated and non-annotated transcripts and a functional approach to identify new antibacterial peptides.

## 2. Results and Discussion

### 2.1. In Silico Analysis of the Transcriptome

The analysis of the transcriptome of the venomous spines and pelvic fins of *Pterois volitans* yielded 47,421 unique transcript sequences. Annotation of the protein sequences using the Uniprot_Swissprot and NCBI NR databases yielded transcripts encoding skin proteins like collagen and keratin, and previously studied venom proteins like toxin α, toxin β and hyaluronidase [53,54]. Other transcripts corresponding to immune factors were also identified, especially four full-length transcripts encoding cysteine-rich antimicrobial peptides: one β-defensin, two peptides related to liver-expressed antimicrobial peptides (hepcidin and LEAP-2), and one NK-lysin. Alignment of lionfish β-defensin with β-defensins of other fish is presented in Figure 1A. The precursor of defensins is organized more simply than the precursors of other cysteine-rich peptides, with a signal peptide followed by 42-amino-acid (AA) β-defensin. The sequence of β-defensin is well conserved among fish, with amino acid identity > 95% with β-defensins of puffer fish (*Tetraodon nigriviridis*), Nile tilapia (*Oreochromis niloticus*) or rainbow trout (*O. mykiss*). The positions of the six cysteines are conserved; only two amino acid residues differ between the β-defensin sequences of *P. volitans*, *T. nigriviridis* and *O. niloticus*. Lionfish β-defensin possesses characteristic features of cationic peptides, with a net charge of 1.7, a molecular weight of 4495.39 Da, and an isoelectric point of 8.35. The occurrence of β-defensin in lionfish pelvic fins was confirmed by mass spectrometry analysis in the alkylated and digested sample using NanoLC-ESI-LTQ-Orbitrap. We detected two trypsic peptides: KVCLPTELFFGPLGCGK and VCLPTELFFGPLGCGK.

Our transcriptomic analysis also identified two precursors encoding peptides related to liver-expressed antimicrobial peptides. The first one was *Pv*-hepcidin. Alignment of the predicted amino acid sequence of *Pv*-hepcidin with hepcidins from other fish is shown in Figure 1B. The highest level of amino acid identity was observed with hepcidins of medaka (*Oryzias melastigma*) (78.99%) and rainbow trout (*O. mykiss*) (77.78%). The predicted prepropeptide of *Pv*-hepcidin contained a 24-AA signal peptide, a 40-AA prodomain, and a 26-AA mature peptide. Sequence variability was mainly observed in the signal peptide and prodomain regions. Between the prodomain region and hepcidin stood a conserved RX(K/R)R motif of propeptide convertase that releases the mature peptide [55]. The predicted mature hepcidin of lionfish contained 8 cysteines with N- and C-terminal sequences conserved with other fish hepcidins. With a calculated molecular weight (MW) of 3001.58 Da, a pI of 8.76, a hydrophobic ratio of 50% and a total net charge of 4, lionfish hepcidin exhibited antimicrobial characteristics. The arrangement of the eight conserved cysteines in hepcidins conferred them the tridimensional structure essential for antimicrobial activity, as described for zebrafish hepcidin [56].

Lionfish hepcidin also possessed the N-terminal sequence QSHL that is essential for iron homeostasis and controls ferroportin internalization and degradation, as observed for zebrafish hepcidin [57]. Lionfish hepcidin, like other vertebrate hepcidins, potentially plays a dual function in iron regulation and host defense. We only found one transcript encoding hepcidin in lionfish, but other genes may be expressed in liver or other tissues, as in several teleosts [7,57,58,59,60]. We also identified a second liver-expressed AMP in lionfish pelvic fin and venomous spines, namely liver-expressed antimicrobial-related peptide 2—*Pv*–LEAP-2. Sequence alignment of *Pv*-LEAP-2 with other LEAP-2 precursors of teleosts highlighted the same conformation, with a signal peptide, a prodomain and a mature peptide (Figure 1C). The conserved RXXR motif for propeptide convertases was located before the cleavage site separating the prodomain from the mature peptide, as observed in other LEAP-2 transcripts [13]. LEAP-2 sequences mainly varied in the prodomain region, whereas the signal peptide and mature peptide sequences were well conserved. The mature peptide *Pv*-LEAP-2 shared high amino acid identity (>90%) with other fish LEAP-2-related peptides. *Pv*-LEAP-2, like other selected fish LEAP-2s, is a 46-AA peptide with four conserved cysteine residues, a net charge of 2, a hydrophobic ratio around 30%, and an isoelectric point ~8.5. While disulfide bonds appear necessary for antimicrobial activity of hepcidins, they do not seem necessary for antimicrobial activity of LEAPs, as suggested for human LEAP-2 [61]. No data are available about the structure-bioactivity relationships in fish. Few antimicrobial activities of fish LEAP-2 have been described. Lionfish LEAP-2 probably has the same activity as *Mm*-LEAP-2, which only differs by four amino acid residues in the C-terminal end [13].

The last cysteine-rich AMP we identified in lionfish was an NK-lysin-related-peptide. The *Pv*-NK-lysin-RP precursor consisted of 153 AA residues. In silico analysis showed that *Pv*-NK-lysin-RP contained a signal peptide sequence (residues 1–22) and a saposin B domain (residues 48–122) well conserved among fish NK-lysins (Figure 1D). Structure prediction of *Pv*-NK-lysin-RP indicated five α-helices spaced out by three loops in the protein molecule, as demonstrated for the structure of porcine NK-lysin and human granulysin, two members of the saposin-like protein family (SAPLIP) [62,63,64]. NK lysins or derived peptides have been showed to display antimicrobial activity. The saposin B domains of three Channel catfish NK-lysins had an antimicrobial effect on *Escherichia coli* and *Staphylococus aureus* [25]. Moreover, antimicrobial activity was observed for sequences corresponding to the cationic core region of the NK lysin saposin domain of *Larimichthys crocea* and *Paralichthys olivaceus* [22,23]. The saposin domain with its six cysteines is well conserved among NK-lysins of fish and lionfish (Figure 1D) and suggests a potential function in the immune response and probable antimicrobial activity.

Only 26% of transcriptome was annotated in Uniprot-Swissprot, 22% in Pfam, leaving more than 34,000 sequences unannotated. To identify other antimicrobial peptides already identified in fish like cathelicidins or other piscicidins but not annotated after similarity search in lionfish transcriptome, we used the software program Peptraq [65]. This in silico search using lionfish transcriptome data identified two new piscidin precursors—pteroicidins B and C. Pteroicidin A had already been identified in a previous study [40]. The precursors of piscidins have very few conserved motifs at the level of the antimicrobial peptide, but their signal peptides are highly similar (Appendix A, Figure A1). Using the FLVL and MAEPG motifs in Peptraq software, two new piscidin precursors were extracted from the database. This confirmed the high sequence homology at the level of the signal peptide. This in silico search also revealed the limitations of BLAST annotation and the usefulness of a software program like Peptraq for targeted searches. It can be a quick way to identify new AMPs in other closely related species, and in particular new piscidins, provided that genomic or transcriptomic databases are available. The study of pteroicidins B and C showed that the sequence corresponding to the signal peptide shared at least 77% identity with those of the other piscidins shown in Figure 1E and F. Pteroicidin C had a high % identity (>55%) with piscidin-4 of white bass *(Morone chrysops),* hybrid striped bass (*Morone chrysops x Morone saxatilis*), striped bass (*Morone saxatilis*)*,* and orange-spotted grouper (*Epinephelus coides)* (Figure 1F), whereas pteroicidin B seemed to be closer to piscidin-7 of *Morone saxatilis* with 48% identity (Figure 1E). No mature forms of piscidin-7 precursors close to pteroicidin were characterized, and no antibacterial activity was demonstrated. Concerning piscidin-4, only one mature peptide was identified from hybrid striped bass, with 44 AA residues and antibacterial activity against fish and human pathogens. The characterization of the precursor of piscidin-4 of hybrid striped bass revealed a four-AA prodomain at the C-terminus [35].

We did not identify any cathelicidins in lionfish. Maybe they are not expressed in the pelvic and dorsal fins, or there is very low sequence homology between lionfish cathelicidins and those of other teleosts. In line with this hypothesis, a cathelicidin identified in Dabry’s sturgeon (*Acipenser dabryanus*) displayed little sequence homology with other fish cathelicidins as compared to hepcidin or defensins [66]. Furthermore, cathelicidin transcripts were found in several tissues including the skin in Dabry’s sturgeon [66], but especially in the spleen and kidney of Atlantic cod [27].

The different transcripts of antimicrobial peptides were expressed in the venomous spines and pelvic fin of *P. volitans* (Figure 2). The most expressed transcript was that of defensin, especially in the pelvic fin. The high expression level of this transcript could explain why only defensin was detected in the pelvic fin by mass spectrometry. The other three transcripts encoding cysteine-rich peptides were very weakly expressed and any peptide has been detected in MS/MS.

Concerning pteroicidins, previously identified pteroicidin A [40] and pteroicidin B were more expressed, especially in the venomous spines. However, MS/MS analyses did not detect mature pteroicidins in the different pelvic fin and venomous spine samples; even α-pteroicidins previously detected in skin [40] was not detected. This result suggests that the antimicrobial peptides derived from these precursors could be released under certain conditions like infection or by other tissues like the gills or the intestine, as described for piscidin-4 of hybrid striped bass [35] or α-pteroicidins in skin tissue [40].

Out of all the AMPs detected by our transcriptomic approach, defensin was the only one identified in all lionfish peptidic samples, and it was only identified in the alkylated, digested sample analyzed by electrospray ionization associated to orbitrap mass spectrometry. This result shows the importance of the mass spectrometer type and of sample preparation protocol, as observed in previous studies [65,67]. Other extraction protocols could be tested on several lionfish tissues to detect mature AMPs.

### 2.2. Study of Peptides Derived from Pteroicidin B and C

The study of antibacterial activity was carried out on peptides that could be encoded by pteroicidins B and C. Cysteine-rich peptides due to the presence of disulfide bridges have a complex three-dimensional structure and are difficult to synthesize, especially on a large scale for use in human or animal health. Therefore, we focused our study on peptides related to piscidins 4 and 7, which are more original and have been little studied.

As we failed to detect the mature forms of the peptides derived from pteroicidins Band C by a peptidomics approach, eight sequences following the signal peptide corresponding to short or long forms (10 to 20 AA) were synthesized. We focused on piscidin-related peptides for the functional approach because piscidins are cysteine-free peptides adopting an α-helix structure, unlike the cysteine-rich peptides identified in this study. Peptides β-Pte20, β-Pte17, β-Pte13 and β- Pte10 were derived from pteroicidin B, and γ-Pte20, γ-Pte17, γ-Pte13 and γ-Pte10 were derived from pteroicidin C. These two series of peptides were very similar in terms of their AA length but also in their physicochemical properties (Table 1). Their hydrophobicity ranged between 40 and 50%, and their net charge varied from +3 to +6, which is commonly observed for cationic AMPs [68].

The peptides were tested on human, fish and oyster pathogens. All minimum inhibitory concentrations (MICs) and minimum bactericidal concentrations (MBCs) are reported in Table 2. The antibacterial spectra were peptide-specific.

The longer sequences β-Pte20, β-Pte17, γ-Pte20 and γ-Pte17 were the most active ones. They were bactericidal on human pathogens but also on the fish pathogen *Aeromonas salmonicida* [69] and on the oyster pathogens *Vibrio aestuarianus* and *V*. *splendidus* [70]. They were bactericidal on all strains except *Enterococcus faecalis*, whose growth was only inhibited by γ-Pte20 and γ-Pte17, with MICs of 1 to 5 μM. While γ-Pte20 and γ-Pte17 had a similar antibacterial effect on most of the tested strains, β-Pte17 often showed higher MICs and MBCs than β-Pte20 did.

The shorter sequences γ-Pte13 and γ-Pte10 (13 and 10 amino acids, respectively) only inhibited *E. coli* and *A. salmonicida* growth. β-Pte10 seemed to be the least active peptide: it inhibited *E. coli* growth only between 25 and 50 μM. β-Pte13 had a bacteriostatic effect on *E. coli* between 1 and 5 μM and a bactericidal effect between 10 and 25 μM (Table 2).

Regarding hemolysis (Figure 3), peptides derived from pteroicidin B and C were very weakly hemolytic at concentrations under 50 µM. They induced less than 5% hemolysis for β-sequences and 10% for γ-sequences. γ-Pte 20 and particularly γ-Pte17 were the most hemolytic peptides at high concentrations (19% and 43% at 200 μM, respectively). The shorter peptides were very weakly hemolytic at all concentrations (≤ 5% hemolysis). The pteroicidin C and B peptides tested in their study had an antibacterial activity profile similar to that observed for α-Pte-NH2 [40]. However, they were less hemolytic, with EC50 values above 200 μM. α-Pte-NH2 caused 100% hemolysis from 50 μM [40].

The structure of an AMP as well as its physical-chemical properties are important because they can modulate its antibacterial and hemolytic activities [59]. In order to understand the different results obtained with pteroidins, we examined the helical wheel conformation of α-Pte, β-Pte and γ-Pte. Twenty-amino-acid peptides like α-Pte revealed a hydrophobic face and seemed to adopt an amphiphilic α-helix. The helical wheel diagrams of β-Pte 20 and γ-Pte 20 (Figure 4) were similar and underlined a cationic face important for interactions with the bacterial membrane, e.g., interactions involving hydrophobicity [71]. Hydrophobicity, the hydrophobic moment and the angle subtended by charged residues modulate the antibacterial and hemolytic activities of peptides [72]. The high hydrophobicity level of α-Pte compared to those of β-Pte 20 and γ-Pte 20 and its low charge can explain its hemolytic activity observed in a previous study [40]. The difference in hydrophobicity and charge observed between γ-Pte-17 (H = 0.414, µ = 0.647, Φ = 5.89, z = 5) and γ-Pte-20 (H = 0.317, µ = 0.556, Φ = 5.99, z = 6) could account for the different hemolysis rates observed at 200 µM.

A circular dichroism analysis was performed on the 13- to 20-AA sequences. Compared to the CD analysis of amidated and non-amidated α-Pte [40] performed in the same conditions, the longest forms (β-Pte20, β-Pte17 and γ-Pte20) seemed to have a very slight α-helix structuring (Figure 5). Trifluoroethanol can mimic an environment conducive to α-helix structuring, but it has certain limitations [73,74]. A medium capturing the properties of bacterial membranes might have allowed better visualization of α-helix structuring.

Peptides β-Pte20 and γ-Pte20 possessed a broad spectrum of antibacterial activity, close to that of piscidins [5], but these peptides or shorter or longer forms were not detected by mass spectrometry. Pteroicidins B and C in their native forms may well be large pteroicidins like piscidins 4, 5, 6 and 7 of hybrid sea bass [34], as the alignments in Figure 1 suggest. Larger peptides are more difficult to identify by mass spectrometry; this could also partly explain why they were not detected by the peptidomic approach. As suggested by the alignments in Figure A2 associated to Table A1, the three lionfish pteroicidins identified in this study and in a previous study [40] appear to correspond to different classes of piscidins, as described for hybrid sea bass piscidins [34]. To confirm this, all endogenous mature forms of pteroicidins will have to be characterized.

This study shows that peptides from the N-terminus of certain pteroicidins can generate antibacterial peptides. This is the case for many other piscidin precursors whose native forms have not been characterized, and yet they are highly active against various pathogens in in vitro tests [75,76,77]. An evolutionary adaptation allowed long piscidins (even in their truncated form) to generate AMPs like histone-derived AMPs [78]. Finally, these peptides are interesting in that they do not induce hemolysis, unlike pteroicidin-α [40], at the concentrations at which they inhibit or are bactericidal against a range of bacteria. This is most likely due to their hydrophobicity, charge and low capacity to structure into α-helices, unlike α-Pte, and also raises questions as to the mode of action of these different peptides, which would be interesting to determine.

## 3. Materials and Methods

### 3.1. Animal Collection

Fish sampling was conducted by spearfishing during SCUBA diving in 2016. Lionfish total length ranged between 19 and 37 cm, corresponding to one-and-a-half to four-and-a-half-year-old animals, respectively, based on the equation proposed by Barbour and collaborators [79]. Dorsal spines and pelvic fins were dissected from twenty lionfish and stored in RNA later solution (ThermoFisher Scientific, Waltham, MA, USA) or 3% acetic acid.

### 3.2. Ethical Statement

*Pterois volitans* is an invading species in the Caribbean Sea. The French Government promotes any sampling able to contribute to the eradication of the species.

### 3.3. Illumina Sequencing

Total RNA was extracted in TriReagent from the dorsal spines or pelvic fins of ten lionfish. The total RNA concentration of each sample was quantified using a NanoDrop spectrophotometer (ThermoFisher), and RNA quality was verified using a bioanalyzer (Agilent Technologies, Waldbronn, Germany). cDNA libraries were prepared with an Illumina TruSeq RNA Sample Preparation Kit v2 (Part# 15008136 Rev. A, Illumina, San Diego, CA, USA) as described in Cornet et al. [80]. The sequencing of 150 paired-end reads was performed on a Miseq system (Illumina).

### 3.4. Bioinformatic Analysis

Two Illumina transcriptomes were obtained from dorsal spines and pelvic fins of lionfish [39], with totals of 8,117,178 and 8,471,617 paired-end reads, respectively, with a maximal read length of 150 bp. Reads were deposited in Sequence Read Archive (SRA) under the accession numbers SRR5141017 and SRR5141018. Adapter removing from the raw reads and quality trimming were performed using Trimmomatic [81]. The transcriptome was assembled using the two libraries and the Trinity platform (https://github.com/trinityrnaseq/trinityrnaseq/releases/tag/Trinity-v2.1.0, accessed on 1 January 2015), a genome-independent transcriptome assembler, for de novo assembly, using the default parameters [82].

Peptide prediction was made using Transdecoder [83]. A similarity search (blastp of the Transdecoder-predicted peptides) was performed against the Uniprot-Swissprot database version 2015_06. Peptide signal prediction was studied using SignalP v4.1 [84]. Transmembrane peptides were detected using TMHMM v2.0c [85]. The protein domain search was performed using hmmscan from the hmmer v.3.1b1 suite against the Pfam-A database release 29.0 [86]. Finally, functional annotation was carried out using the Trinotate pipeline (http://trinotate.github.io, accessed on 1 January 2015) described by Haas and collaborators [83].

To complete the analysis and search for peptides with low sequence homology such as piscidins, we used the homemade software program PepTraq [65,87,88]. The strategy developed to search for new piscidins was to use the sequence homologies observed at the signal peptide level of the precursors encoding piscidins. These sequence homologies have been described in previous studies [89,90] and correspond to the FLVL and MAEPG patterns.

Additionally, molecular weights (MW) and isoelectric points (pI) were calculated using the Expasy compute pI/Mw tool (http://web.expasy.org/compute_pi/, accessed on 1 January 2015), and multiple sequence alignment was performed using CLC Main Workbench6 (CLC BIO). Domain prediction was performed with SMART (http://smart.embl-heidelberg.de, accessed on 1 January 2015).

Cleaned reads were remapped for each library independently on the full transcriptome using Bowtie2 [91], and relative abundance values were estimated using RSEM (http://deweylab.github.io/RSEM, accessed 1 January 2018, [92]) to get FPKM (fragments per kilobase of exon per million fragments mapped) values.

PepTraq software was used for in silico analysis. Peptraq was developed to perform in silico analyses based on “-omic” data (https://peptraq.greyc.fr, accessed on 1 May 2021). The search for precursors or peptides through PepTraq can be achieved using several structural criteria, as described by Zatylny-Gaudin and collaborators [65] for neuropeptides or by Houyvet and collaborators [87] or Benoist and collaborators [92] for antibacterial peptides.

### 3.5. Peptidomic Analysis

One gram of dorsal spines or pelvic fin obtained from five lionfish was homogenized in acetic acid 3% at a 1:10 (*w*/*v*) ratio and centrifuged 20 min at 35,000× *g* at 4 °C. The supernatants were concentrated on Sep-pak C18 cartridges (Waters, Milford, MA, USA) and dried in a speed-vacuum. Each sample was reduced with 100 mM DTT at 55 °C for 60 min, and alkylated with 50 mM iodoacetamide at 55 °C for 45 min. Half of the alkylated and reduced samples were hydrolyzed with trypsin at 25 ng/μL at 37 °C to recover larger peptides. The resulting peptides were analyzed by NanoLC-MALDI-TOF/TOF and NanoLC-ESI-LTQ-Orbitrap as described in [65].

### 3.6. Selection and Synthesis of Peptides from Pteroicidins B and C

Eight peptides were selected from the precursors of pteroicidins B and C. For each of them, the first 20 amino acids located after the cleavage site of the signal peptide constituted the parent peptides, which were the longest peptides. These peptides were subdivided into 3 truncated forms of 17, 13 and 10 amino acids according to the potential monobasic cleavage sites (see Table 3). The physico-chemical properties including the charge, and molar mass of each of these peptides were determined with the software program available online from the Antimicrobial Peptide Database https://aps.unmc.edu/, accessed on 1 August 2003) [93] and Heliquest. The helical wheel projection diagrams of peptides were predicted using Heliquest [94] (http://heliq uest.ipmc.cnrs.fr/cgibin/ComputParamsV2.py, accessed on 1 July 2008).

### 3.7. Antimicrobial Assay

The antibacterial activity of peptides β-Pte20, β-Pte17, β-Pte13 and β-Pte10 from pteroicidin B and γ-Pte20, γ-Pte17, γ-Pte13 and γ-Pte10 from pteroicidin C was evaluated on several bacteria as presented in Table 3. All peptides were solubilized in sterilized water. The minimum inhibitory concentrations (MICs) were determined according to the liquid growth inhibition test following the protocol established by Hetru and Bulet [95] and used for the study of α-pteroicidins [40]. Briefly, 10 μL of peptide solution were added to 100 μL of bacterial suspension at a starting OD_600_ of 0.001 (~10^5^ to 10^6^ CFU/mL according to strain) and incubated in 96-well microplates. The culture media of the different bacteria are listed in Table 1. Bacterial growth was assessed by optical density measurement at 595 nm after 16 h of incubation. All the tests were performed three times, and each peptide concentration between 0.1 and 50 µM was tested in triplicate. MICs were expressed in µM as an [a]–[b] concentration interval where [a] was the last concentration with bacterial growth and [b] was the first concentration with 100% bacterial growth inhibition. MBCs (minimal bactericidal concentrations) were expressed in µM as an [a]–[b] concentration interval where [a] was the last concentration with bacterial growth and [b] was the first concentration that killed 100% of the bacteria, determined after the plating on agar plates [95].

### 3.8. Hemolytic Assay

The hemolytic activity of peptides β-Pte20, β-Pte17, β-Pte13 and β-Pte10 from pteroicidin B and γ-Pte20, γ-Pte17, γ-Pte13 and γ-Pte10 from pteroicidin C was evaluated on human red blood cells (RBCs) following the protocol previously described in [96] and used for α-Pte in a previous study [40]. RBCs were obtained from nine blood samples from French blood donors thanks to the support of the Etablissement Français du Sang (EFS). RBCs were rinsed twice in PBS and then diluted to obtain a 1% erythrocyte solution (~5 × 10^6^ cells determined using a Scepter™ 2.0 cell counter). Ninety μL of erythrocyte solution were incubated with 90 μL of peptide solution at concentrations ranging from 5 to 200 μM, for one hour at 37 °C without shaking. The percentage of hemolysis was determined by optical density measurement at 415 nm. The experiment was performed in triplicate. The 0% hemolysis and 100% hemolysis controls were determined in PBS and 1% Triton X-100, respectively.

### 3.9. Structural Circular Dichroism Study

The structural properties of peptides β-Pte20, β-Pte17, β-Pte13 and β-Pte10 from pteroicidin B and γ-Pte20, γ-Pte17, γ-Pte13 and γ-Pte10 from pteroicidin C were evaluated by circular dichroism. The peptides were studied at a concentration of 0.1 mg/mL diluted in PBS buffer or in 35% and 75% trifluoroethanol (TFE). CD spectra were obtained using a J810 spectrometer (JASCO, Woonsocket, RI, USA) in a 1-mm long cell. The CD spectra and the helix contents were analyzed by the method developed by Micsonai and co-workers [97] and the software program BESTSEL available online https://bestsel.elte.hu/ accessed on 1 January 2015.

## 4. Conclusions

The present study reports for the first time the identification of four cysteine-rich antimicrobial peptides in the Scorpaenidae family: hepcidin, LEAP-2-RP, NK-lysin-RP, and β-defensin. No cathelicidin was identified. Two new pteroicidin precursors-B and C—were identified using conserved motifs within the signal-peptide-coding sequence of piscidin precursors. The discovery of these two precursors in the lionfish shows that lionfish has three categories of piscidins, like other teleosts., Eight peptides of 10 to 20 amino acids were synthesized from pteroicidins B and C. Functional tests showed that whatever the original precursor, the longest forms were the most active ones on the different bacteria tested including human, fish and oyster pathogens. These non-hemolytic antibacterial peptides could constitute an alternative to antibiotics for humans but also for aquatic organisms present in aquaculture facilities, particularly in hatcheries.

## Figures and Tables

**Figure 1 marinedrugs-19-00490-f001:**
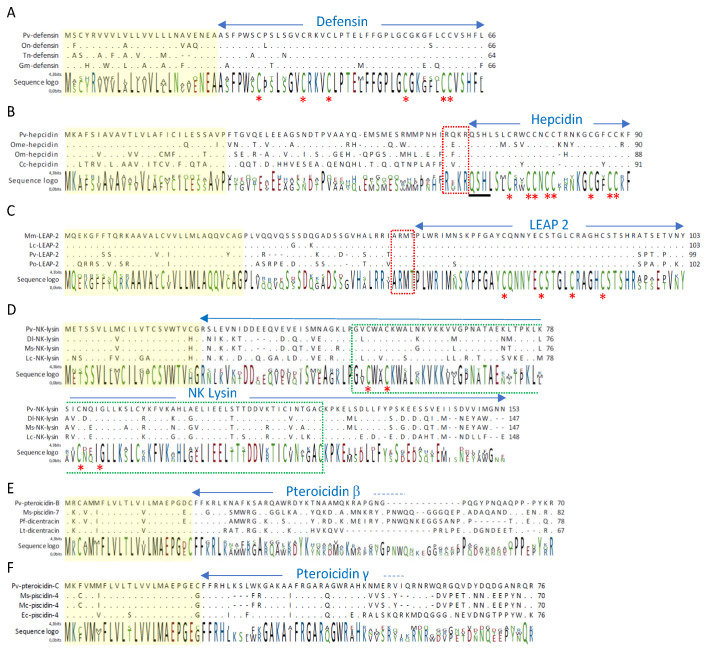
Sequence alignment of antimicrobial peptide precursors identified in lionfish. (**A**) Alignment of *Pterois volitans* Pv-defensin with *Tetraodon nigroviridis* Tn-defensin (CAJ57644.1), *Oreochromis niloticus* On-defensin (AGW83444.1), and *Gadus morhua* Gm-defensin (AEB69787.1). (**B**) Alignment of *P. volitans* Pv-hepcidin with *Oryzias melastigma* Ome-hepcidin (AEG78327.1), *Oncorhynchus mykiss* Om-hepcidin (ADU85830.1) and *Cyprinus carpio* Cc-hepcidin (AFY23859.1). Dotted red boxes, RX(K/R)R motif for cleavage of the mature hepcidin peptide; underlining, QSHL motif. (**C**) Alignment of *P. volitans* Pv-LEAP-2 (liver-expressed antimicrobial peptide 2) with *Miichthys miiuy* Mm-LEAP-2 (AHN13905.1), *Larimichthys crocea* Lc-LEAP-2 (AFC90192.1), and *Paralichthys olivaceus* Po-LEAP-2 (ACB97648.1). Dotted red boxes, RXXR motif for the cleavage of the mature LEAP-2 peptide. (**D**) Alignment of *P. volitans* Pv-NK-lysin with *P. olivaceus* Po-NK-lysin (AU260449.1) and *L. crocea* Lc-NK-lysin (AIL25791.1). Dotted blue box, saposin B domain; red asterisks, cysteines. (**E**) Alignment of *P. volitans* Pv-pteroicidin-B with *Morone saxatilis* Ms-piscidin-7 (KX231322.1), *Perca flavescens* Pf-dicentracin (XP_028460572.1), *Liparis tanakae* Lt-dicentracin (TNN22976.1). (**F**) Alignment of *P. volitans* Pv-pteroicidin-C with *M. saxatilis* Ms-piscidin-4 (APQ32049.1), hybrid [*M. chrysops x M. saxatilis*] Mh-piscidin-4 (ADP37959.1), and *Epinephelus coioides* Ec-piscidin 4 (AKA60777.2). The sequence logo summarizes the consensus sequence and amino acid variations with fish sequences. Yellow, signal peptide; Blue arrow, antimicrobial sequence.

**Figure 2 marinedrugs-19-00490-f002:**
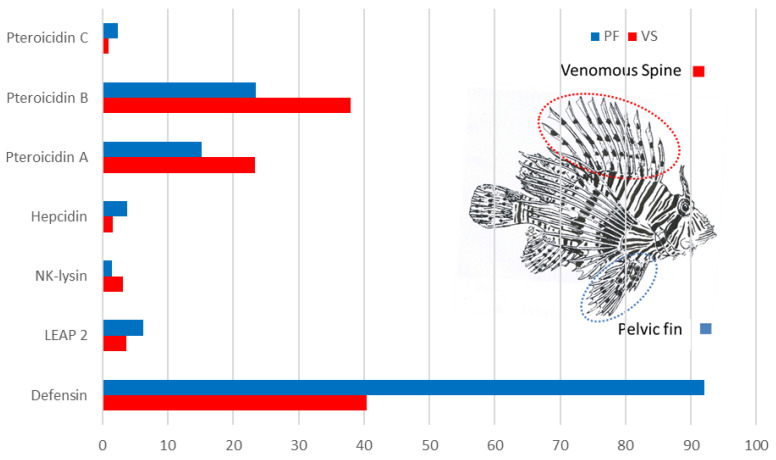
Expression of the different antimicrobial peptide transcripts identified in the venomous spines (red) and pelvic fin (blue) of lionfish. (FPKM = Fragments Per Kilobase of exon per Million fragments mapped).

**Figure 3 marinedrugs-19-00490-f003:**
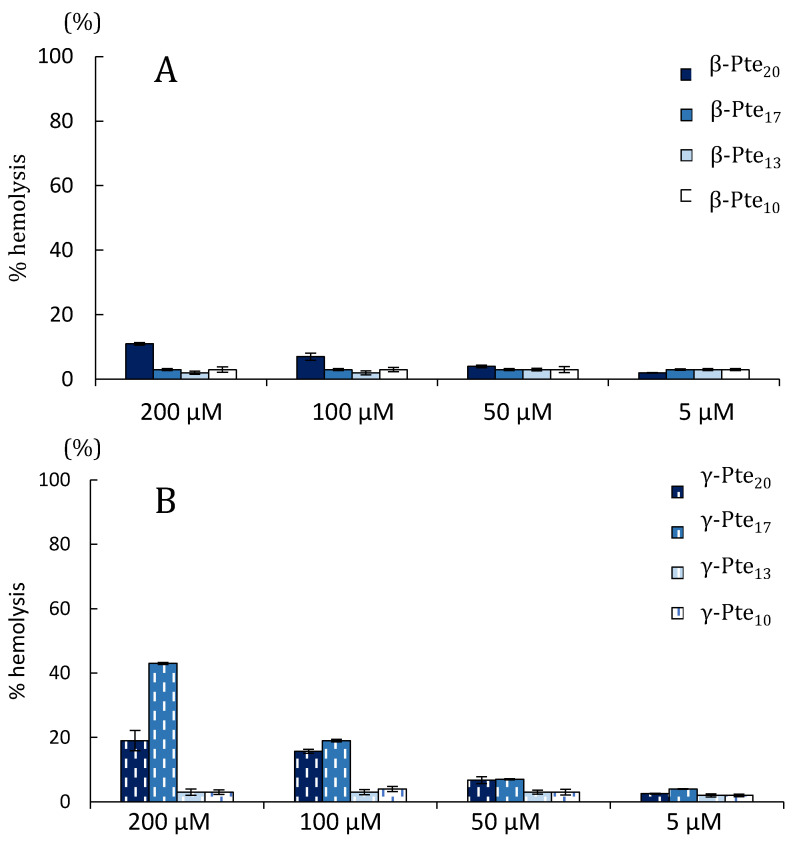
Hemolytic activity (%) of the synthetic peptides. (**A**) β-Pte20, β-Pte17, β-Pte13 and β-Pte10; (**B**) γ-Pte20, γ-Pte17, γ-Pte13 and γ-Pte10 (**B**), Hemolytic activity was evaluated at different concentrations on human erythrocytes.

**Figure 4 marinedrugs-19-00490-f004:**
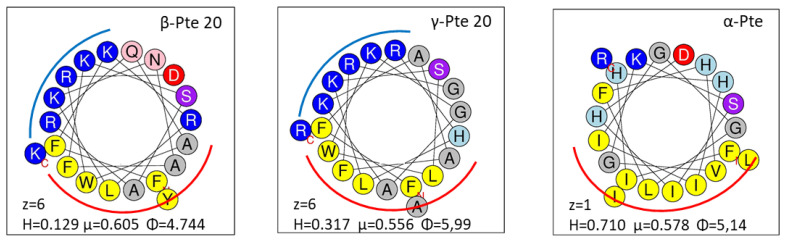
Helical wheel diagram of β-Pte 20, γ-Pte 20 and α-Pte. The helical wheel projections were performed using the HeliQuest online program: https://heliquest.ipmc.cnrs.fr/cgi-bin/ComputParams.py (accessed on 1 July 2008). Yellow, hydrophobic amino acids; blue, basic residues; red, acidic residues; purple and pink, polar amino acids. The red line indicates hydrophobic face, the blue line, cationic face. z, charge; hydrophobicity (H), hydrophobic moment (μ) and angle (Φ).

**Figure 5 marinedrugs-19-00490-f005:**
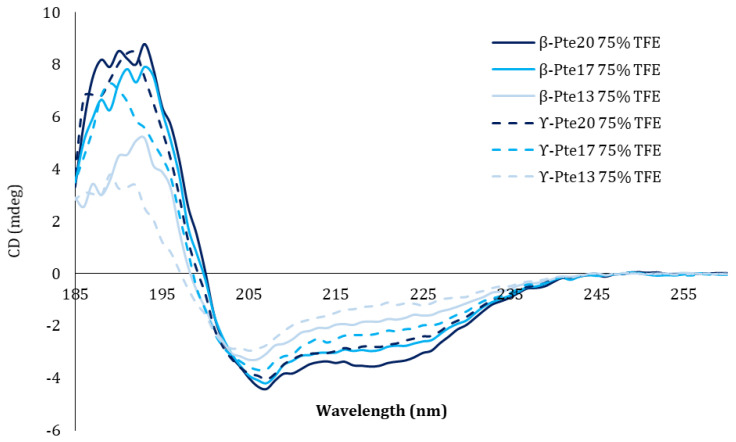
Circular dichroism analysis. Circular dichroism monitoring of the structuring of the peptides synthesized from pteroicidin B and pteroicidin C precursors in the presence of 75% trifluoroethanol (TFE).

**Table 1 marinedrugs-19-00490-t001:** Physicochemical properties of the peptides selected from pteroicidins B and C.

Peptide	Amino Acid Sequence	Number of AA	MW (Da)	Net Charge	Hydrophobicity
β-Pte20	FFKRLKNAFKSARQAWRDYK	20	2561	6	40%
β-Pte17	FFKRLKNAFKSARQAWR	17	2155	6	47%
β-Pte13	FFKRLKNAFKSAR	13	1613	5	46%
β-Pte10	FFKRLKNAFK	10	1299	4	50%
γ-Pte20	FFRHLKSLWKGAKAAFRGAR	20	2348	6	50%
γ-Pte17	FFRHLKSLWKGAKAAFR	17	2063	5	52%
γ-Pte13	FFRHLKSLWKGAK	13	1618	4	46%
γ-Pte10	FFRHLKSLWK	10	1362	3	50%

**Table 2 marinedrugs-19-00490-t002:** Antibacterial activity of the peptides synthesized from pteroicidin B and C precursors in µM.

Bacteria		Peptides from Pteroicidin B							
		β-Pte20		β-Pte17		β-Pte13		β-Pte10	
		MIC	MBC	MIC	MBC	MIC	MBC	MIC	MBC
Gram-positive	*Listeria monocytogenes*	1–5	5–10	5–10	25–50	NA	ND	NA	ND
	*Enterococcus faecalis*	NA	ND	NA	NA	NA	ND	NA	ND
	*Staphylococcus aureus*	1–5	10–25	10–25	25–50	NA	ND	NA	ND
Gram-negative	*Escherichia coli*	1–5	5–10	1–5	1–5	1–5	10–25	25–50	ND
	*Salmonella typhimurium*	10–25	10–25	10–25	10–25	NA	ND	NA	ND
	*Aeromonas salmonicida*	1–5	1–5	5–10	5–10	NA	ND	NA	ND
	*Vibrio aesturianus*	1–5	5–10	10–25	10–25	NA	ND	NA	ND
	*Vibrio splendidus*	1–5	5–10	10–25	10–25	NA	ND	NA	ND
	*Vibrio vulnificus*	5–10	5–10	10–25	10–25	NA	ND	NA	ND
Bacteria		**Peptides from Pteroicidin C**							
		γ-Pte20		γ-Pte17		γ-Pte13		γ-Pte10	
		MIC	MBC	MIC	MBC	MIC	MBC	MIC	MBC
Gram-positive	*Listeria monocytogenes*	1–5	5–10	1–5	5–10	NA	ND	NA	ND
	*Enterococcus faecalis*	1–5	NA	1–5	NA	NA	ND	NA	ND
	*Staphylococcus aureus*	1–5	1–5	1–5	1–5	NA	ND	NA	ND
Gram-negative	*Escherichia coli*	1–5	1–5	1–5	10–25	10–25	10–25	25–50	25–50
	*Salmonella typhimurium*	5–10	5-10	5–10	5–10	NA	ND	NA	ND
	*Aeromonas salmonicida*	1–5	1–5	1–5	1–5	25–50	NA	25–50	NA
	*Vibrio aestuarianus*	5–10	10–25	5–10	5–10	NA	ND	NA	ND
	*Vibrio splendidus*	5–10	5–10	10–25	10–25	NA	ND	NA	ND
	*Vibrio vulnificus*	10–25	10–25	10–25	10–25	NA	ND	NA	ND

MIC, minimum inhibitory concentration; MBC, minimum bactericidal concentration; NA, non active; ND, not determined.

**Table 3 marinedrugs-19-00490-t003:** List of the bacterial strains used in this study.

	Bacterial Strain	Reference Number	Culture Medium	Temperature
Gram-positive	*Listeria monocytogenes*	CIP 110871	BHI	37
	*Enterococcus faecalis*	CIP 76.117	LB	37
	*Staphylococcus aureus*	CIP 53.1 56	CL	37
Gram-negative	*Escherichia coli*	CIP 54.8T	LB	37
	*Salmonella typhimurium*	CIP 103446	TSB	37
	*Aeromonas salmonicida*	CIP 103209T	CL	30
	*Vibrio aestuarianus*	CIP 109791T	MB	25
	*Vibrio splendidus LGP32*	CIP 107715	MB	25
	*Vibrio vulnificus*	CIP 109783	MB	30

CIP: Collection of the Institut Pasteur. MB: Marine Broth (Conda): 40.20 g/L, pH 7.6; LB, Luria-Bertani: peptone 10 g/L, yeast extract 5 g/L, NaCI 10 g/L, pH 7.5; BHI, Brain Heart Infusion (Difco): 37 g/L, pH 7.4; TSB, Trypticasein Soy Broth (Conda): 30 g/L, pH 7.3; CL, Columbia broth (Conda): 35 g/L, pH 7.4.

## Data Availability

https://www.ncbi.nlm.nih.gov/bioproject/?term=prjna360033 (accessed on 1 January 2015); https://www.ncbi.nlm.nih.gov/bioproject/?term=prjna360035 (accessed on 1 January 2015).

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
