# Peer review of "Marine Transcriptomics Analysis for the Identification of New Antimicrobial Peptides"

_marinedrugs, 2021, doi:10.3390/md19090490_

Round 1

Reviewer 1 Report

The manuscript describes the identification of AMPs in lionfish (Pterois volitans). The topic is interesting and actual, but the manuscript needs to be improved before publishing. The authors do not report Materials & Methods well, which hampers the reproduction of their experiments and increases reporting bias. They also do not provide all data. Some parts of the manuscript are not well written and should be rewritten.

Specific issues:

  1. “Antimicrobial peptides have been studied in many teleosts [40], but only one study was carried out a venomous teleost_ lionfish (Pterois volitans)- led to the identification of pteroicidin a, a piscidin-like-peptide.” (P2/L95-97) The sentence should be reformulated.
  2. I would avoid “data not shown” (P3/L112, P3/127). The authors should report all results avoiding reporting bias.
  3. I do not understand what the authors mean by this sentence “The firstAMP was Pv- β-defensin.” The first AMP in what?
  4. “Its high expression level could explain its detection by mass spectrometry.” (P3/223-224) This sentence does not make sense.
  5. Table 2 – It is not clear why some numbers are bold.
  6. Two paragraphs reporting the antibacterial and hemolytic activities of AMPs (P8/L253-268) are not easy to follow. I recommend rewriting them to be more readable and understandable.
  7. I do not understand what the authors want to say by this sentence: “Peptides β-Pte20 and γ-Pte20 possessed a broad spectrum of antibacterial activity, close to that of piscidins [40], but our data do not allow us to conclude about their biological facts.” (P10/L283-285) The authors should say what they mean by stating that the biological activity of β-Pte20 and γ-Pte20 is close to that of piscidins. Which biological facts can not be concluded?
  8. Peptides β-Pte20 and γ-Pte20 were hemolytic at higher concentrations; their selectivity was 10x max. The authors compare their biological activities to alpha-Pte without providing more data. The authors should give antibacterial and hemolytic activities and selectivity of a-Pte in their experimental set-ups to directly compare these AMPs.
  9. The authors should provide more information about used fish (age, weight, sex).
  10. The authors should spell out the abbreviation SRA - Sequence Read Archive.
  11. The authors mention the peptidomics approach & mass spectrometry analysis (P3/L127, P6/L223-224, P10/287-289). However, they do not describe them in the Material & Methods, provide the results, and discuss them. They should do all.
  12. The authors should state the solvent used for the studied AMPs. Was solvent control included? If not, it should be. The authors should provide the source of bacterial strains and specify them. They should also note CFU/mL used as bacterial inoculum for each bacterial strain. The culture media are not specified in Table 1 (P12/L370-371). The authors should describe how they determined MIC and MBC. How bactericidal activity was measured is not even described. The description of the used methods is not detailed to be repeated by somebody else. The authors should improve that. How many times were these experiments repeated independently? They should do it at least three times.
  13. The composition of culture media and temperature can affect in vitro kinetics of AMPs (free available concentrations, binding and so on) and subsequently their real potency to inhibit bacterial growth, killing bacteria or lysing red blood cells. The authors should discuss it in their manuscript.
  14. The authors should state the source of human red blood cells and describe the donors. They should specify how many cells were exposed to AMPs. Was optical density measured in the presence of red blood cells? How many times were these experiments repeated independently? They should do it at least three times.

Author Response

Dear reviewer

Thank you for yours comments that allowed us to improve the quality of the manuscript.

Please find the new version of our manuscript with corrections in blue.

A new figure has been added ( figure 4), and a table ( table 3).

The responses to your comments have been added below in blue.

1. “Antimicrobial peptides have been studied in many teleosts [40], but only one study was carried out a venomous teleost_ lionfish (Pterois volitans)- led to the identification of pteroicidin a, a piscidin-like-peptide.” (P2/L95-97) The sentence should be reformulated.

The sentence has been modified (L95-97).

2. I would avoid “data not shown” (P3/L112, P3/127). The authors should report all results avoiding reporting bias.

We completed these parts L114-116. Bibliographical references have been added. The peptidomic approach has been described (L129-132, L244-255, L408-416).

3. I do not understand what the authors mean by this sentence “The firstAMP was Pv- β-defensin.” The first AMP in what? In lionfish.

The sentence has been removed (L120).

4. “Its high expression level could explain its detection by mass spectrometry.” (P3/223-224) This sentence does not make sense.

The sentences L 238 -240 have been modified.

5. Table 2 – It is not clear why some numbers are bold.

Table 2 has been modified, bold numbers have been removed.

6. Two paragraphs reporting the antibacterial and hemolytic activities of AMPs (P8/L253-268) are not easy to follow. I recommend rewriting them to be more readable and understandable.

The paragraphs have been modified (L275-299).

7. I do not understand what the authors want to say by this sentence: “Peptides β-Pte20 and γ-Pte20 possessed a broad spectrum of antibacterial activity, close to that of piscidins [40], but our data do not allow us to conclude about their biological facts.” (P10/L283-285) The authors should say what they mean by stating that the biological activity of β-Pte20 and γ-Pte20 is close to that of piscidins. Which biological facts can not be concluded?

The sentences L 336-337 have been modified to clarify this point.

8. Peptides β-Pte20 and γ-Pte20 were hemolytic at higher concentrations; their selectivity was 10x max. The authors compare their biological activities to alpha-Pte without providing more data. The authors should give antibacterial and hemolytic activities and selectivity of a-Pte in their experimental set-ups to directly compare these AMPs.

The Materials and Methods section has been completed (L445-446), and paragraphs modified (L292-299).

9. The authors should provide more information about used fish (age, weight, sex).

The size of the fish were between 19 and 37 cm (L358). Weight varied a lot depending on predation. Age was estimated (L359 -360).

10. The authors should spell out the abbreviation SRA - Sequence Read Archive.

Sentence L 375 have been modified

11. The authors mention the peptidomics approach & mass spectrometry analysis (P3/L127, P6/L223-224, P10/287-289). However, they do not describe them in the Material & Methods, provide the results, and discuss them. They should do all.

The peptidomic approach with mass spectrometry analysis has been added and the Discussion has been modified, as indicated previously (L129-132, L244-255, L408-416).

12. The authors should state the solvent used for the studied AMPs. Was solvent control included? If not, it should be. The authors should provide the source of bacterial strains and specify them. They should also note CFU/mL used as bacterial inoculum for each bacterial strain. The culture media are not specified in Table 1 (P12/L370-371). The authors should describe how they determined MIC and MBC. How bactericidal activity was measured is not even described. The description of the used methods is not detailed to be repeated by somebody else. The authors should improve that. How many times were these experiments repeated independently? They should do it at least three times.

The AMPs are dissolved in water and we have control growth with water.

Protocols and methods have been detailed in previous works.

A description of the strains has been added (Table 3).

All the tests were performed three times, and each peptide concentration was tested in triplicate.

Paragraph 3.7 has been modified (L428-441).

13. The composition of culture media and temperature can affect in vitro kinetics of AMPs (free available concentrations, binding and so on) and subsequently their real potency to inhibit bacterial growth, killing bacteria or lysing red blood cells. The authors should discuss it in their manuscript.

It is difficult to discuss these points. No tests with temperature or media changes were performed. All antibacterial and hemolytic tests were performed under standard conditions well described previously.

14. The authors should state the source of human red blood cells and describe the donors. They should specify how many cells were exposed to AMPs. Was optical density measured in the presence of red blood cells? How many times were these experiments repeated independently? They should do it at least three times.

Red blood cells came from nine blood samples from French blood donors thanks to the support provided by the EFS (Etablissement Français du Sang). Hemolytic tests were performed under standard conditions well described previously.

Three tests were performed independently.

The Materials and Methods section has been modified (L445-447).

Concerning English language, the article was proof-read for English language by Annie Buchwalter, a professional translator.

Thank you for your expertise

sincerely yours,

Céline Zatylny-Gaudin

Reviewer 2 Report

The authors describe mainly a bioinformatical approach to characterize the transcriptome of the lionfish (Pterois volitans) with special focus to the antimicrobial peptide fraction in venomous spines and pelvic fins of this fish. The in silico analysis presented seems to be sound and shows sequence data of most important AMP families in the lionfish which then have been compared to other yet sequenced AMP data from other fish species (Figure 1). The expression intensity of each AMP family was determined (Fig. 2). From the pteroicidin family A and B each 4 peptide sequences (Table 1) have been selected, synthesized and tested on gram-positive and -negative bacteria (Table 2), including analyzing their hemolytic (Figure 3) and potential structural features by CD spectroscopy (Figure 4).

Although interesting, however this study can be substantially optimized.

  1. In the introduction it is stated that the success of lionfish invasion maybe e.g. due to the lack of microbial pathogens. I agree to this hypothesis but in the manuscript no work was spent on this interesting aspect. Is there anything known about microbial fish pathogens in the Carribean compared to the microbial fish pathogens in south-east asia where the lionfish originally is living ?
  2. Under this concept such specific bacterial strains could be tested, if possible, to see if this hyopthesis is right or not, rather then general human pathogenic bacteria tested here (Table 2).
  3. In Figure 1 the sequences of Pv-derived AMP sequences are compared to other fish-derived AMP sequences showing significant and expected sequence homology in most cases. The alignement is sound for me. However sequencing the transcriptome will not reveal the real processed and active peptide sequences that are present in the lionfish. This confirmation has to be experimentally carried out by high-sensitive mass spectrometry or LC-MS of the samples. The authors mention that some MS data have been generated and confirmed e.g. the existence of defensins in lionfish. It would largely enhance the significance of this manuscript if at least for some exemplary cases the existance of Pv-AMPs can be confirmed by peptide MS data. This data also will provide information of the processed forms of AMPs in the lionfish. So to my opinion the peptidomic data mentioned (line 288/289) should be added.
  4. The authors state (chapter 2.2.), that the 8 AMP sequences for antimicrobial testing have been selected due to pragmatical reasons (size; easy to synthesize). So its not clear or more or less unlikely that exactly these peptides are present in the lionfish. Of course, the sequences are lionfish specific. However, if these peptides represent valuable sequences for putative development of antibiotic drugs as stated in the text, remains open. Maybe testing on some multidrug resistant bacteria could support this hypothesis. Or testing on fish-relevant bacteria of fungi would be more meaningful for this study.
  5. The CD experimental data are not convincing. Why didn´t the authors measure the peptides in water and in water with 10% SDS. This would be much more closer to the physiological conditions (and mimicking some membrane environment) rather then to use 75% Trifluoroethanol. In addition to Figure 4 the secondary structure content in percentage has to be added which can be easily calculated by programs like bestsel.elte etc.

Author Response

Dear reviewer,

Thank you for your comments  that allowed us to improve the quality of the manuscript.

Please find the new version of our manuscript with corrections.

A new figure (figure 4) and a new table (table 3) have been added 

The responses to the comments have been added below in blue.

1. In the introduction it is stated that the success of lionfish invasion maybe e.g. due to the lack of microbial pathogens. I agree to this hypothesis but in the manuscript no work was spent on this interesting aspect. Is there anything known about microbial fish pathogens in the Carribean compared to the microbial fish pathogens in south-east asia where the lionfish originally is living ?

The Introduction has been modified. (L103-107). Bibliographical data on this subject is scarce. No lionfish pathogens have been described in the wild or in captivity in aquaria.

2. Under this concept such specific bacterial strains could be tested, if possible, to see if this hyopthesis is right or not, rather then general human pathogenic bacteria tested here (Table 2).

No lionfish pathogens have been described. We tested fish and oyster pathogens equally. Table 2 has been modified, and the Discussion too (L275-288).

3. In Figure 1 the sequences of Pv-derived AMP sequences are compared to other fish-derived AMP sequences showing significant and expected sequence homology in most cases. The alignement is sound for me. However sequencing the transcriptome will not reveal the real processed and active peptide sequences that are present in the lionfish. This confirmation has to be experimentally carried out by high-sensitive mass spectrometry or LC-MS of the samples. The authors mention that some MS data have been generated and confirmed e.g. the existence of defensins in lionfish. It would largely enhance the significance of this manuscript if at least for some exemplary cases the existance of Pv-AMPs can be confirmed by peptide MS data. This data also will provide information of the processed forms of AMPs in the lionfish. So to my opinion the peptidomic data mentioned (line 288/289) should be added.

We thoroughly agree, mass spectrometry analysis has been added, and the Discussion has been modified (L129-132, L244-255, L408-415).

4. The authors state (chapter 2.2.), that the 8 AMP sequences for antimicrobial testing have been selected due to pragmatical reasons (size; easy to synthesize). So its not clear or more or less unlikely that exactly these peptides are present in the lionfish. Of course, the sequences are lionfish specific. However, if these peptides represent valuable sequences for putative development of antibiotic drugs as stated in the text, remains open. Maybe testing on some multidrug resistant bacteria could support this hypothesis. Or testing on fish-relevant bacteria of fungi would be more meaningful for this study.

We tested fish and oyster pathogens equally. Table 2 has been modified, and the Discussion too. We agree with the reviewer that other tests on other pathogens could be interesting however, considering the short time-lapse to perform corrections to the article, we were not able to perform the new assays.

5. The CD experimental data are not convincing. Why didn´t the authors measure the peptides in water and in water with 10% SDS. This would be much more closer to the physiological conditions (and mimicking some membrane environment) rather then to use 75% Trifluoroethanol. In addition to Figure 4 the secondary structure content in percentage has to be added which can be easily calculated by programs like bestsel.elte etc.

CD performed in other conditions is indeed needed to confirm peptide structure, we completed this part by adding a comparison with α-Pte and an analysis with heliquest (L300-331).

Concerning English language, the article was proof-read for English language by Annie Buchwalter, a professional translator.

Thank you for your expertise,

sincerely yours

Céline Zatylny-Gaudin

Round 2

Reviewer 1 Report

The authors addressed most of the comments. However, I have still some major issues:

  1. "While the peptides had antibacterial activities close to that observed for α-Pte-NH2, their hemolytic activity was very different: although α-Pte-NH2 and α-Pte were both highly hemolytic, 100% hemolysis was reached from 50 μM for α-Pte-NH2 versus 100 μM for α-Pte [40]." (P8/L296-299) This sentence does not make sense. The authors should reformulate it to say that Pteroicidin C and B-peptides tested in their study had an antibacterial activity profile similar to that observed for α-Pte-NH2 [40]. However, they were less hemolytic with EC50 above 200 μM. α-Pte-NH2 caused 100% hemolysis at 50 μM and higher [40].
  2. The authors should specify CFU/mL (not just stated OD600) used as bacterial inoculum for each bacterial strain.
  3. The authors did not explain how MIC and MBC were determined. How did they confirm that 100% of bacteria were killed? Just based OD600? If so, they should verify that by the plating on agar plates.
  4. The authors should specify how many red blood cells were exposed to AMPs, not just stated 1% erythrocyte solution
  5. What does it mean "One Health context" in the Conclusion?

Author Response

We thank the reviewer for his/her expertise. However, in view of the time given by the editor it is difficult to give all the expected information.

The answers to each comment are in green below, as well as the new modifications of the manuscript.

We hope that the new corrections will be suitable.

Sincerely Yours,

Céline Zatylny-Gaudin

1. "While the peptides had antibacterial activities close to that observed for α-Pte-NH2, their hemolytic activity was very different: although α-Pte-NH2 and α-Pte were both highly hemolytic, 100% hemolysis was reached from 50 μM for α-Pte-NH2 versus 100 μM for α-Pte [40]." (P8/L296-299) This sentence does not make sense. The authors should reformulate it to say that Pteroicidin C and B-peptides tested in their study had an antibacterial activity profile similar to that observed for α-Pte-NH2 [40]. However, they were less hemolytic with EC50 above 200 μM. α-Pte-NH2 caused 100% hemolysis at 50 μM and higher [40].

The sentences have been modified (L 297-299).

2. The authors should specify CFU/mL (not just stated OD600) used as bacterial inoculum for each bacterial strain.

The sentence has been modified (L.465-437)

Liquid antibacterial tests use optical density at 600 nm  as described in several studies, not CFU or CFU/mL which vary according to each bacterium .

We chose to start the test with the same OD for all bacteria.

3. The authors did not explain how MIC and MBC were determined. How did they confirm that 100% of bacteria were killed? Just based OD600? If so, they should verify that by the plating on agar plates.

This information is cited in the added references.

The instructions to authors indicate that only new protocols should be fully described.

Bactericidal effect has been determined by the plating on agar plates.

To satisfy reviewer 1, this information has been added to the Materials and Methods section (L440-445) .

5. The authors should specify how many red blood cells were exposed to AMPs, not just stated 1% erythrocyte solution

This was an oversight, please excuse us. The information has been added.

6. What does it mean "One Health context" in the Conclusion?

The sentence has been modified.

These non-hemolytic antibacterial peptides could constitute an alternative to antibiotics for humans but also for aquatic organisms present in aquaculture facilities, particularly in hatcheries.

Reviewer 2 Report

The manuscript has improved considerably. I would recommend to publish after careful English revision.

Round 3

Reviewer 1 Report

The authors have addressed all my concerns.